# Improved Biohythane Production from Rice Straw in an Integrated Anaerobic Bioreactor under Thermophilic Conditions

**DOI:** 10.3390/microorganisms11020474

**Published:** 2023-02-14

**Authors:** Lili Dong, Guangli Cao, Wanqing Wang, Geng Luo, Fei Yang, Nanqi Ren

**Affiliations:** 1Key Laboratory of Agro-Forestry Environmental Processes and Ecological Regulation, School of Ecology and Environment, Hainan University, Haikou 570208, China; 2State Key Laboratory of Urban Water Resources and Environment, School of Environment, Harbin Institute of Technology, Harbin 150090, China

**Keywords:** rice straw, continuously biohythane production, integrated reaction bioreactor, NaOH/Urea pretreatment, thermophilic fermentation

## Abstract

This study evaluated the feasibility of continuous biohythane production from rice straw (RS) using an integrated anaerobic bioreactor (IABR) at thermophilic conditions. NaOH/Urea solution was employed as a pretreatment method to enhance and improve biohythane production. Results showed that the maximum specific biohythane yield was 612.5 mL/g VS, including 104.1 mL/g VS for H_2_ and 508.4 mL/g VS for CH_4_, which was 31.3% higher than the control RS operation stage. The maximum total chemical oxygen demand (COD) removal stabilized at about 86.8%. COD distribution results indicated that 2% of the total COD (in the feed) was converted into H_2_, 85.4% was converted to CH_4_, and 12.6% was retained in the effluent. Furthermore, carbon distribution analysis demonstrated that H_2_ production only diverted a small part of carbon, and most of the carbon flowed to the CH_4_ fermentation process. Upon further energy conversion analysis, the maximum value was 166.7%, 31.7 times and 12.8% higher than a single H_2_ and CH_4_ production process. This study provides a new perspective on lignocellulose-to-biofuel recovery.

## 1. Introduction

In recent years, climate change and energy production scenarios have emerged as a global challenge and increasingly unanimously appealed to action by the international community [1,2]. According to a literature, the global energy demand will increase by 35% until 2035 [3]. In China alone, until 2020, the total energy consumption amount reached 4980 million tons of oil equivalent—58% of primary energy consumption—leading to both energy and environment crises [4]. It has been well recognized that developing clean-burning and renewable energy can replace fossil energy consumption. In a comparison of petrol/natural gas, hydrogen (H_2_) is a clean energy carrier due to its significant properties, such as a broad range of applications, a high calorific value (142 KJ/g), and being CO_2_-neutral and renewable [5,6]. Methane (CH_4_) is another crucial energy carrier widely used in the chemical industry due to its various commercial values [2,7]. A new valuable energy carrier, hythane (mixture of H_2_ and CH_4_), is characterized by a H_2_/CH_4_ ratio (5–20%) suitable for improving combustion engine performance and environmental impact. In addition, it has been reported that the higher the ratio of H_2_/CH_4_, the better the quality of the hythane [2,8]. Thus, the suggested hydrogen content in hythane is 10–25% by volume [9].

Of all the hythane production routes, two-stage anaerobic digestion (AD) biological lignocellulosic biomass waste conversion to hythane offers a long-term potential for sustainable hythane production with environmental friendliness. It is widely regarded as one of the most promising hythane production routes in the future [10,11], because biohythane is an advanced biofuel that can improve energy recovery and reduce atmospheric stress. Rice straw (RS) is a type of agricultural lignocellulosic waste with an annual yield of 6.7 × 10^8^ t/a worldwide [12,13]. From the perspective of bioenergy recovery and waste management, the most convenient method of remediation of RS is using microbes for biofuel production. However, due to the chemical composition and recalcitrance structure of RS, raw RS is unapproachable for direct biohythane production. Thus, prior to the biological hythane conversion process, pretreatment is needed to remove the lignocellulose recalcitrance and improve its biodegradability and enzyme accessibility [5]. Our previous study demonstrated that NaOH/urea (NU) solution pretreatment at outdoor cold-winter conditions could effectively enhance RS’s enzymatic saccharification and hythane production [13]. Hence, RS was pretreated by this pretreatment and was used as the substrate for biohythane production in this study.

Continuous biohythane production via an integrated system has been nominated to improve the economy by operating two independent bioreactors in financial, energy, human resources, and commercialization issues [2,8,14,15]. Many studies have been focused on soluble organic substrates, different reactor coupling forms, fermentation temperature, and so on [8,9,15,16,17]. Recently, a liquid of cellulosic biomass or microcrystalline cellulose has also been used as a substrate for H_2_ and CH_4_ production [18]. It has been reported that anaerobic fermenting at a moderate thermophilic temperature (50–65 °C) provided higher H_2_ and CH_4_ yields than mesophilic conditions (25–40 °C). Continuous biohythane production from sugarcane stillage using an acidogenic anaerobic fluidized bed reactor (AFBR-A) and a methanogenic AFBR (AFBR-S) was studied by Ramos and Silva (2020); the highest biohydrogen yield in the continuous stirred tank reactor (CSTR) and methane yield in the AFBR were 7.6 (±1.8) mL H_2_/g COD_added_ and 0.26 (±0.06) L CH_4_/g COD [19]. However, few studies have been conducted on the direct utilization of lignocellulose biomass substrates for biohythane fermentation [2,13]. One of the main obstacles was delivering the lignocellulose biomass to the lab-scale reactors. Furthermore, most of the studies have been carried out in batch mode. To our best knowledge, no reports are on the directly biohythane production from lignocellulose biomass, especially when carried in an integrated anaerobic bioreactor (IABR), or exploring the performance and carbon distribution of the whole biohythane production process.

The main aim of this study was to investigate continuous biohythane production potentials and stability from RS in an IABR at a lab scale for 50 days; the substrate was RS pretreated by NU, and RS without pretreatment was used as a control group. Firstly, critical parameters of the IABR start-up and operation process, fermentation substrate concentration, and hydraulic retention time (HRT) were explored using batch experiments. Then, the performance and stability of RS conversation in IABR were investigated. During the bioreactor operation process, daily biogas production, gas composition, and the chemical oxygen demand (COD) were determined to prove the effectiveness of the performance of continuous biohythane production from RS. Finally, carbon distribution, energy conversion, and flow throughout the whole biohythane production were evaluated and analyzed. This study is the first to employ RS to direct biohythane production in IABR. Still, this study offers important information for lignocellulose biomass to high-value biohythane production.

## 2. Materials and Methods

### 2.1. Substrate and Inoculum Characteristics

#### 2.1.1. Substrate

The experiment’s substrate was RS, taken from Wuchang, Heilongjiang province, China. RS’s total solid (TS) and volatile solid (VS) were 98.1% and 88%. Prior to use, RS was air-dried and ground then passed through a 40-mesh sieve. Based on dry matter, the compositions of raw RS were 35.97% cellulose, 27.45% hemicellulose, and 14.08% lignin, respectively. Pretreated RS was selected for our previous study, which was pretreated by 3%–6% NU at 100% solid loading for 3 months at outdoor cold-winter conditions, whose composition was 46.98% cellulose, 29.13% hemicellulose, and 5.88% lignin [13]. Meanwhile, the VS of pretreated RS was 98%.

#### 2.1.2. Inoculum

The H_2_- and CH_4_-producing inoculum used in this research was collected from a plug flow reactor (PFR), which used RS as substrates for CH_4_ production and had been operating for more than three months under thermophilic conditions. The parameters of biogas slurry were as follows: the pH and oxidation–reduction potential (ORP) were 7.15 ± 0.02 and −313 ±9.21. The TS, VS, COD, and volatile fatty acids (VFAs) were 0.97 ± 0.23, 0.82 ± 0.13, and 9640 mg/L, and 124.85 ± 10.21 mg/L, respectively. The biogas slurry needs to be activated in a constant temperature incubator at 60 °C to eliminate the influence of the background substrate before inoculating. A vital step was to aerate the biogas slurry until the dissolved oxygen was below 3 mg/L. Then, it was used as the inoculum for H_2_ production and added at 30% (*v/v*). The mixed microflora was added at 100% (*v/v*), which was used as the inoculum for CH_4_ production.

### 2.2. Batch Biohythane Production Tests for Optimizing Substrate Concentration

Substrate concentration tests for biohythane production from RS were carried out in 250 mL anaerobic serum bottles with a working volume of 50 mL. The RS before and after pretreatment was used as carbon source and set as 10 g/L (1% total solid (TS)), 20 g/L (2%TS), and 30 g/L (3%TS)) to produce hythane by two-stage AD. The composition and sterilization of the fermentation medium were described by Dong et al. 2018 [5]. In H_2_ production stage, 30% (*v/v*) of pretreated biogas slurry was inoculated into the medium. After 72 h fermentation, 80% (*v/v*) of biogas slurry was inoculated into the medium for the CH_4_ fermentation, and this stage continued until 144 h. The fermentation process was cultivated at 60 °C. During the whole fermentation process, the quantity and compositions of produced gas were measured every 24 h over 144h. All experiments were carried out in triplicate to check data reproducibility.

### 2.3. Reactor Configuration and Operation

#### 2.3.1. Reactor Configuration

An IABR (Figure 1) for continuous AD RS was constructed with transparent Plexiglass sheets. It is ten-millimeter-thick. The IABR had a capacity of 5.84 L and a working volume of 5.4 L. The H_2_ production (HPA) area was 1 L with an active working volume of 0.9 L. The CH_4_ production (MPA) area was 4.84 L with a dynamic working volume of 4.5 L, which includes two transition areas before and after MPA with a capacity of 0.5 L and a dynamic working volume of 0.45 L. Another, there was 0.44 L for gas space. Magnetic stirrers were installed in both the HPA and MPA areas, and the rotating speed was set at 170 rpm. Temperature measuring, sampling, and gas collection ports were at the top IABR. The sampling port of HPA was on the side of the bioreactor, which was simple 1 and 3. The sampling port of MPA was placed on the top of the bioreactor, which was simple 2 and 4. The external IABR was surrounded by a tropical belt to keep the constant temperature required (60 °C) for the bioreactor.

#### 2.3.2. Reactor Start-Up and Operation

The IABR was operated under 1% TS (equivalent to about 10,000 mg/L COD). Before the start-up, 100% (*v/v*) treated biogas slurry was added to the bioreactor, and the aerated biogas slurry was added to the HPA area. The aim was to provide an anaerobic environment for microorganisms. A total of 900 mL fermentation medium (contained 9 g RS) with 30% biogas slurry was input into the bioreactor on the first day, and there was no feeding on the second day. In other words, the organic load rate (OLR) is equivalent to 10 g/(L day) in the whole operation process. Dong et al. (2018) described the fermentation medium, and the carbon resource was RS. In the follow days, 450 mL of the preconfigured substrate was continuously fed into the reactor by pumping. The reactor’s pressure was balanced by the new feed and gas production through the gas collection ports on the top of the reactor. The material liquid after digestion was collected by an effluent tank. The whole operation process can be divided into three different phases: start-up (from 1 to 3 days), control operation phase (RS was untreated) (4–34 days), and operation phase (NU pretreated RS) (35–50 days).

#### 2.3.3. Energy Conversion Efficiency (ECE) Calculation

The energy conversion efficiency (ECE) of this work was calculated based on the equation shown below [5]:$$\mathrm{ECE}\%=\frac{\mathrm{Heat}\;\mathrm{value}\;\mathrm{of}\;H_{2}\;\left(\mathrm{kJ}\right)}{\mathrm{Heat}\;\mathrm{value}\;\mathrm{of}\;\mathrm{rice}\;\mathrm{straw}\;\left(\mathrm{kJ}\right)}\times 100\%$$

The specific heat values (HV) of H_2_, CH_4_, and RS were 142 kJ g^−1^, 89.05 kJ g^−1^, and 17.6 kJ g^−1^ [20,21,22].

### 2.4. Analytical Methods

During the operation, the sampling of inflow and outflow from HPA and MPA was taken every day. Then, the pH, COD, VFAs, and gas compositions (H_2_, CH_4_, and CO_2_) were analyzed. The COD was measured by HACHHQ40d multi and HACH-DRB 200. The produced gas was collected by gas bags and measured by the dewatering method reversible cycle and registration. The gas products (H_2_, CH_4_, and CO_2_) and VFAs were determined by gas chromatography (GC, 7890A, Agilent Cooperation, Santa Clara, CA, USA) [5]. Soluble polysaccharides were detected by high-performance liquid chromatography (HPLC) system (LC-10A, Shimadzu Corporation, Kyoto, Japan).

## 3. Results and Discussion

### 3.1. Effect of Substrate Concentration on Biohythane Production

To realize the effect of substrate concentration on the biohythane production process, 10 g/L (1% TS), 20 g/L (2% TS), and 30 g/L (3% TS) were investigated in a two-stage fermentation process. Figure 2 shows the performance of biohythane production from RS before and after NU pretreatment, especially the performance of the H_2_ production stage. As can be observed in Figure 2a, Figure 2c, the trend of gas production at different substrate concentrations was the same. The difference was the yield of gas production. As depicted in Figure 2b, H_2_ was released after 12 h inoculation, and the volume gas production yield was obtained as 1.6 L/L, 1.28 L/L, and 0.96 L/L under the substrate concentrations of 1% TS, 2% TS, and 3% TS after 24 h inoculation, respectively. The gas production rate was 1.02 L/(L day), 0.72 L/(L day), and 0.88 L/(L day). With the fermentation processing, the efficiency of gas production decreased with the increase in substrate concentration, especially when the substrate concentration was up to 3% TS, and the efficiency of gas production dramatically reduced. Moreover, the H_2_ production rate decreased with the increase in substrate concentration: 0.33 L/(L day), 0.29 L/(L day), and 0.288 L/(L day). After 72 h fermentation, the specific yields of 1% TS, 2% TS, and 3% TS were 52.52 mL/g TS (58.35 mL/g VS), 28.87 mL/g TS (32.08 mL/g VS), and 21.5 mL/g TS (23.89 mL/g VS). On the one hand, the process of H_2_ production was inhibited at a high substrate concentration in this study. Different factors have been explored to enhance the dark fermentation process of lignocellulosic biomass to improve the H_2_ production efficiency, including pretreatments, fermentation conditions, reactor configuration, microbial strain, or a combination of the preceding methods [2,11,16]. Among these, the substrate concentration is one of the crucial factors affecting a bioenergy-generating system’s productivity, especially biohydrogen production [23]. In addition, as reported by Ntaikou et al. (2010) [24], substrate types, a drop in pH, an increase in the hydrogen partial pressure, or the accumulation of a toxic substance and other factors can direct or indirectly inhibit the H_2_ producing rate and/or the final yield. After 72 h fermentation, H_2_ was no longer detected. On the contrary, gas production significantly increased with biogas slurry inoculation, and CH_4_ started to be detected. The results suggested that the fermentation process was methanogenic fermentation from 72 h. It can be seen from Figure 2a, Figure 2c that the CH_4_ production continued with some fluctuations. Moreover, it was noticed that 1% TS fermentation concentration displayed more effective utilization (Figure 2d), and the substrate was pretreated by NU at outdoor cold-winter conditions. The volume gas production rate was obtained as 1.72 L/L after 48 h fermentation, and the content of H_2_ was 27.1%, which improved 68.6% in comparison of untreated RS. The gas and H_2_ production rates were 1.1 L/(L day) and 0.44 L/(L day). The specific H_2_ yield was 99.28 mL/g TS (101.31 mL/g VS) after 72 h fermentation. This value improved by 89% compared with untreated RS at 1% TS. The results shown in Figure 2d could also demonstrate that NU pretreatment at outdoor cold-winter conditions was an effective way to improve biohydrogen production from RS. Our previous study established that NU pretreatment operating directly outdoors in cold winter was a practical and feasible chemical way to enhance energy production from lignocellulose [13]. Thus, 1% TS (10 g/L RS) and 48h were selected as the substrate concentration and HRT for HPA. The results also indicated that the HRT of MPA should be more than 4d. Thus, combined with bath fermentation results, the HRT of MPA was selected as 10 days in the next experiment. 

### 3.2. Performance of Biohythane Production from Rice Straw

#### 3.2.1. Continuous Gas Production

Gas production and quantification are two critical indicators for substrate’s conversation efficiency and biohythane system stability. Figure 3a shows daily gas production and volume gas production. Concerning components of the produced gas, H_2_ and CO_2_ were the two main components, and CH_4_ and CO_2_ were in MPA (Figure 3b). During the control operation stage, only an average of 130 mL/d gas production was obtained during the first 3 days in HPA. A lag or adaptation period in the bacteria was the main reason, when it was acclimatized to the new environment [13,25]. Similar to gas production, the average H_2_ content was only 10.6%. From day 4 onwards, an increasing gas production trend of HPA was observed. At day 34, this value was stable, at around 247.3 mL/d with 18.2% H_2_ content. The gas production initially increased slowly in MPA, while a rapid increase was found after about 10 d. From day 12 onwards, the gas production was relatively stable, at around 2357 mL/d, with some fluctuations. Remarkably, almost no CH_4_ was detected from day 1 to day 4, indicating that methanogenic bacteria have a negative CH_4_ production process. This fact is because methanogens have a more extended generations to adapt to a new fermentation environment and substrate type. Moreover, in the first two days, batch feeding method was adopted in HPA, resulting in no substrate influencing MPA. Then, CH_4_ content varied from 15% to 48.1%, with a significant fluctuation between day 5 and day 20, reflecting that the CH_4_ production process was unstable during this operation stage. It also indicated that methanogens took longer to reach a normal state. After day 21, the variation in the average CH_4_ percentages in the gas was 64.9% until day 34. This result indicated that the methanogenesis process was in a normal state in MPA. Similarly, Yang et al. (2015) observed stability in an anaerobic digester when the CH_4_ content was 53–70%. Thus, the total daily gas production was 2604.3 mL with 1.7% H_2_ and 58.7% CH_4_ [26].

During the operation phase, the substrate was NU-pretreated. To further obtain the performance of producing biohythane directly from RS, from day 36 onwards, the daily gas production significantly increased in HPA, and the peak value was 521 mL/d on day 43. H_2_ percentages were also up to 32.2% with an average of 30.3%, which was 12.23% higher than the control operation phase. These results demonstrated that NU pretreatment could effectively enhance the bioconversion performance of lignocellulosic biomass. Afterwards, gas production stayed around 475.7 mL/d with 30.3% H_2_ percentages until the IABR stopped operation. This result showed an 89.9% increase in gas production compared with the control operation phase. Still, the average daily gas production of MPA was 2265 mL/d, which was only a 3.9% decrease compared with the control operation phase. However, the CH_4_ content was 75.4% from day 35 to the operation end, which was 10.5% marginally higher than the control operation phase. This is important for improving the quality of hythane. A higher H_2_ and CH_4_ content in gas mean more industrial application value [2]. Moreover, total daily gas production was 2740.7 mL with 5.3% H_2_ and 62.3% CH_4_ during this operation stage. Usually, hythane is a mixture of H_2_ (5–25% by volume) and CH_4_ (75–95% by volume) [7]. Therefore, the gas produced from pretreated RS was more suitable for the quality of hythane. Still, the yield of gas production increased by 5.2%.

#### 3.2.2. Specific H_2_, CH_4_, and Biohythane Production Yield

Figure 4 depicted the variation in specific H_2_ and CH_4_ production yield per unit TS in the case of the control and operation stage. From day 1 to day 5, the average H_2_ production in the HPA was only 42.2 mL/d, and the average specific H_2_ production yield was only 16.4 mL/g VS. Meanwhile, the total CH_4_ production in MPA was only 18 mL, with a specific CH_4_ production yield of 4.3 mL/g VS. From day 6 onwards to day 34, H_2_ and CH_4_ production significantly improved. The peak value of 136.1 mL/d with a specific H_2_ production yield of 75.9 mL/g VS was reached on the 29th day, and the average specific H_2_ production yield was 55.6 mL/g VS. Meanwhile, a very satisfactory average CH_4_ production was 1764.9 mL/d, with a specific CH_4_ production yield of 410.8 mL/g VS. Thus, the average specific biohythane production yield was 466.4 mL/g VS during the control operation stage. From day 35 to the operation’s end, it could be noted that the daily H_2_ production gradually increased to 280.5 mL/d on the 43rd day, with a maximum specific H_2_ production yield of 104.1 mL/g VS. In the following days, the daily H_2_ production and specific H_2_ production yield were observed to be stable, at 248.1 mL/d and 96.2 mL/g VS. However, CH_4_ production had almost no rise. The maximum CH_4_ production was 1932 mL/d, with a specific CH_4_ production of 508.4 mL/g VS. Moreover, the average and CH_4_ production were 1708.8 mL/d and 459.1 mL/g VS. Hence, the average specific biohythane yield of the operation stage was recorded as 555.3 mL/g VS. Clearly, the maximum specific biohythane yield of the operation stage was 612.5 mL/g VS, which increased 31.3% compared with the control operation stage.

#### 3.2.3. COD Removal in Biohythane Production Process

Figure 5 shows the variation in COD removal efficiency of the HPA and MPA in the biohythane production process. For the first day of start-up, there was a high level of COD, with 10,500 mg/L, due to the full amount of substrate being input in the medium. From day 3 to the end of the operation, the daily influent COD was around 10,000 mg/L, with small fluctuations. The COD removal efficiency overall for the whole process was found in a range of 1.12–3.55%, with 2.04% average removal efficiency. This was due to most COD transforming into volatile acid and ethanol, resulting in a lower removal rate of COD in HPA. On the contrary, the degradation rate of TCOD in MPA was close to 100% at the initial stage, while this value gradually decreased to 78.1% in the 9th. This demonstrated that the digestion process was proper at the initial days in the MPA, so a high removal yield of COD was achieved. From day 9 to day 34, the COD removal efficiency retained around 72.3%. However, the removal yield of COD improved from day 35, reaching a maximum of 88.97% at the 38th. Moreover, the removal yield of COD stabilized at about 86.8% until the reactor ended operation. In our previous study, COD removal could stabilize at approximately 70.56% when the carbon sources of AD were fibrous substrates [25]. Furthermore, a 51–79% COD removal rate was also proved by Rowse (2011). As such, these results show that the performance of MPA had been in its stationary state in the operation stage [27].

### 3.3. Carbon Distribution Analysis during Biohythane Production

The carbon in the solid-phase matrix was hydrolyzed into the liquid phase by hydrolytic acidifying bacteria during biohythane production. Meanwhile, H_2_ and CO_2_ and other liquid-phase by-products were generated. Furthermore, liquid-phase metabolites were further converted into CH_4_ and CO_2_. On the other hand, the carbon in RS was transformed into carbon-containing metabolites (soluble polysaccharides, VFAs, CH_4_, and CO_2_) via the process of producing acid and methane, so the analysis of carbon distribution can provide an overview of the transformation process of substrate. The distribution of carbon was calculated and tracked based on all carbon-containing matter. The soluble and gaseous carbon distribution in the operation stage is shown in Figure 6. Carbon accounted for 76.2% of VFAs and CO_2_ when soluble polysaccharides were continuously converted into H_2_ in HPA. This suggested that the carbon source was derived from the continuous hydrolysis of solid matter in HPA. Still, soluble polysaccharides provided a part of the matrix, accounting for 21.7%. Then, the liquid metabolites of HPA could serve as the carbon source of MPA for CH_4_ production. As such, carbon accounted for 87% of CH_4_ and CO_2_ in MPA, especially in CH_4_. These results were seen to be in line with the data obtained from biohythane production performance. Moreover, the solid residues were still continuously hydrolyzed in MPA, and were not completely utilized in HPA. These hydrolysates also served as the part of substrate for MPA. This result was confirmed by the 1.6-fold consumption of VFA carbon content when CH_4_ was produced in MPA. Thus, H_2_ production in this study only diverted a small part of carbon, and most of the carbon flowed to the CH_4_ fermentation process. Concerning the total COD, only 2% (in the feed) was converted into H_2_; 85.4% was converted to CH_4_ during the whole biohythane production process. Still, 12.6% was retained in the discharge, including microorganisms and the residual utilized solid phase. Although, the calculation and analysis of carbon distribution in this study are relatively macroscopic, generalities indeed reflected the overall trend. Moreover, it is more suitable for research on a large-scale program and the acquisition of trends.

### 3.4. Prospects and Challenges of Biohythane Production from Straw

Production of biohythane directly and indirectly from cellulosic biomass wastes has been a widely discussed topic [28]. In 2009, the commercialization of dark fermentative H_2_ production technology had been optimistically anticipated by Urbaniec and Grabarczyk [29]. Sapporo Breweries Limited intended to build a precommercial plant for hythane production from wastes of vegetables and crops in 2013 [30]. However, no more information about the biohythane factory was available. For now, this vision has to be postponed due to the low-cost pretreatment technologies and technological processes limiting the transfer from laboratory-scale to industrial application. The pretreatment method is the critical part, due to it being a highly costly and energy-intensive stage. Although chemical pretreatments are effective and direct methods, they have a main obstacle restricting their large-scale commercial and industrial applications. This issue involves many unavoidable microbe-unfriendly inhibitors such as 5-HMF, phenolic, and furfural compounds during the pretreatment process [31]. Moreover, at the end of the pretreatment, a large amount of water is required in the washing process. Compared to chemical pretreatment, biological pretreatment methods are relatively mild but need a longer processing time and efficient microorganisms, and are affected by environmental factors. Accordingly, it is worth thinking about developing a better solution based on the perspective of using and recycling waste. Table 1 lists several publications of the relevant work. It can be seen that the H_2_ and CH_4_ production performance of this work would be comparable to—or even better than—using other fermentation conditions. Furthermore, compared with the relevant studies, the total biohythane production in this study increased by 37~89%. These results further revealed that biohythane production from NU-pretreated RS via IABR was successful and satisfactory.

Furthermore, energy analysis can effectively be employed to evaluate the overall biohydrogen and biomethane by AD [5,27,28]. According to the results of the previous fermentation, it can be clarified that the ECE of the control operation stage was 148.5%, with 3.9% in HPA and 144.6% in MPA, whereas when the substrate was NU-pretreated RS, an ECE of 166.7% was obtained, 15.3% higher than the control operation. Remarkably, it was also noticed that the ECE of the biohythane process was 31.7 times and 12.8% higher than a single H_2_ and CH_4_ production process. It implies that biohythane is an economically attractive route for lignocellulose to biofuels. In addition, the ECE result greatly depends on the substrate type and composition. Thus, comparing the results with others described in the literature is difficult. Nevertheless, the realization of this paper straw-based NHU pretreatment for biohythane production could provide clean energy by renewable biomass and waste resource integration and recycling. Similarly, it is also a double-benefit clean energy solution and form of environmental protection.

## 4. Conclusions

This study demonstrated that continuous biohythane production from RS was feasible and satisfactory. NU pretreatment had a profound impact on the performance of H_2_ production and the quality of hythane. The maximum specific yield of biohythane was 612.5 mL/g VS with specific H_2_ and CH_4_ of 104.1 mL/g VS and 508.4 mL/g VS, which was 31.3% higher than the control without pretreatment. The maximum COD removal stabilized at about 86.8%, corresponding to H_2_ production from HPA accounting for 2% of the total COD, while CH_4_ production from MPA accounted for 85.4%. The maximum energy conversion efficiency of 166.7% could be achieved 31.7 times and 12.8% higher than a single H_2_ and CH_4_ production process. This work successfully improved biohythane production from IABR with RS and provided helpful information on the lignocellulose-to-hythane routes.

## Figures and Tables

**Figure 1 microorganisms-11-00474-f001:**
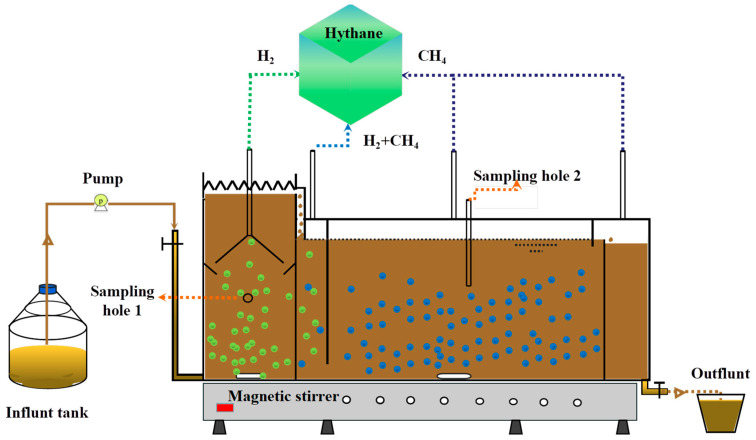
The setup of integrated anaerobic bioreaction (IABR) system for biohythane production.

**Figure 2 microorganisms-11-00474-f002:**
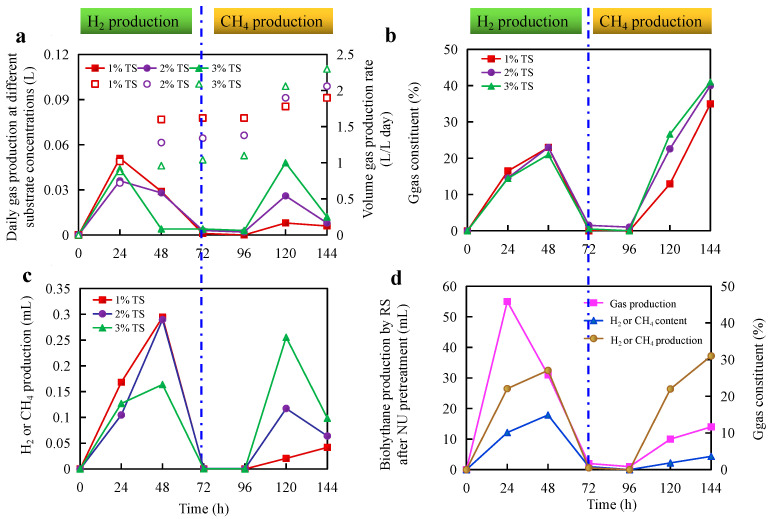
Optimization of substrate concentration of biohythane production: (**a**) gas production performance of untreated rice straw; (**b**) the content of H_2_ and CH_4_ in gas; (**c**) H_2_ and CH_4_ production of untreated rice straw; (**d**) the H_2_ and CH_4_ production performance of rice straw by NaOH/urea pretreatment at 100% solid loading under outdoor cold winter conditions for 3 months at 1% substrate concentration.

**Figure 3 microorganisms-11-00474-f003:**
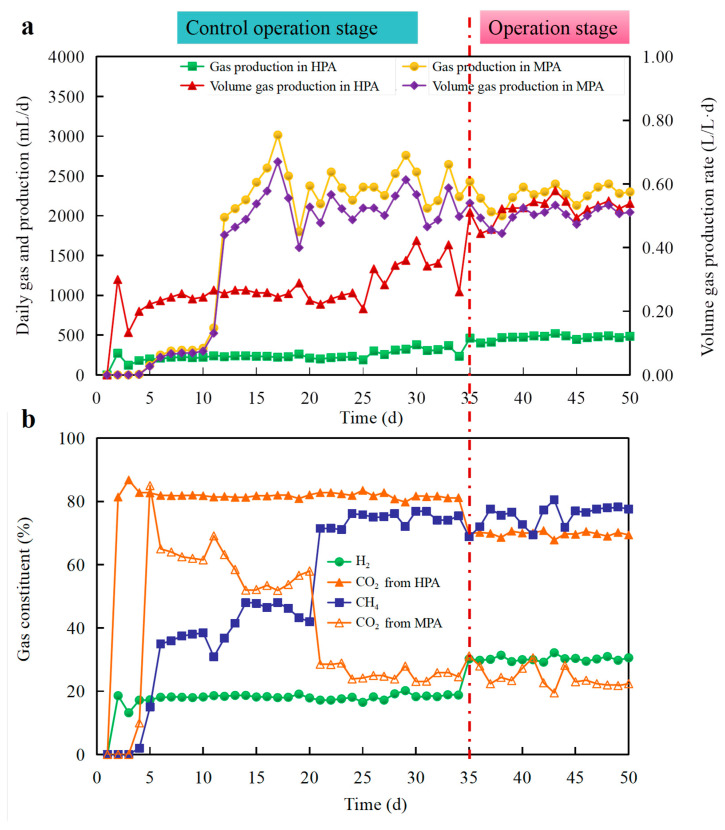
Performance of biohythane production from rice straw. (**a**) Continuous gas production; (**b**) variation in H_2_, CH_4_, and CO_2_ content in produced gas.

**Figure 4 microorganisms-11-00474-f004:**
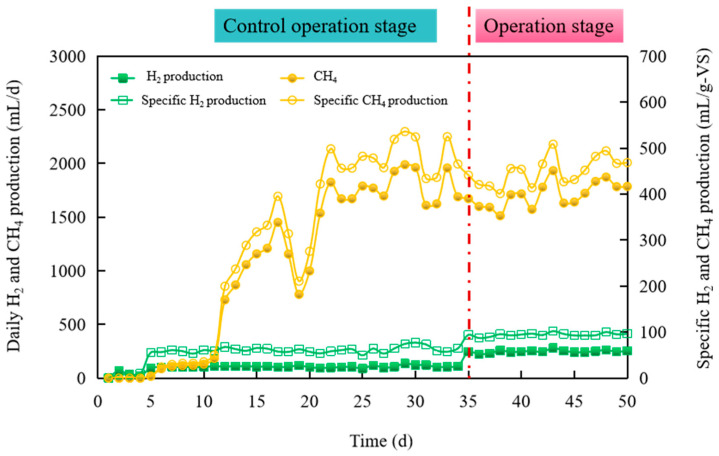
The variation in specific H_2_ and CH_4_ production yield during different operation stages and areas.

**Figure 5 microorganisms-11-00474-f005:**
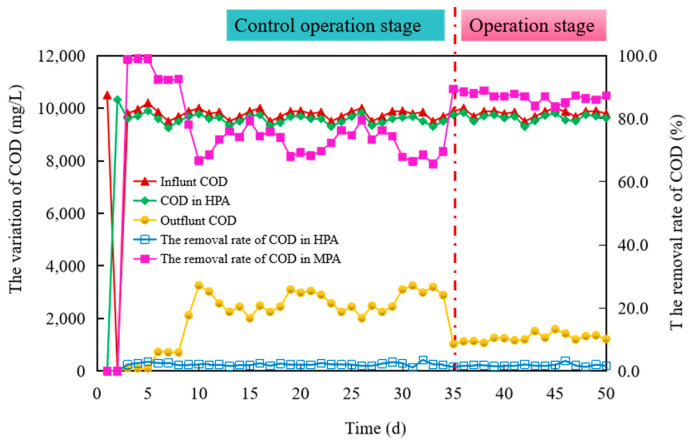
Variation in the chemical oxygen demand (COD) and removal rate during biohythane production process.

**Figure 6 microorganisms-11-00474-f006:**
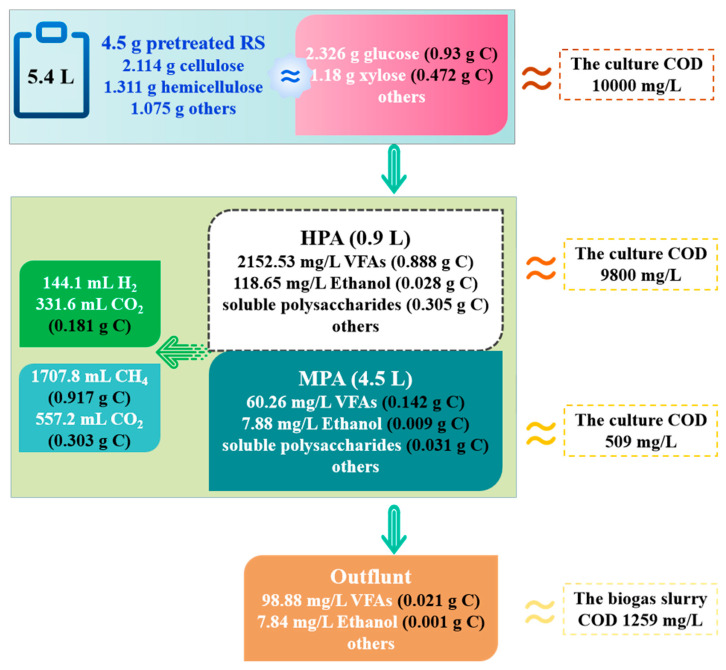
The carbon distribution during biohythane production process.

**Table 1 microorganisms-11-00474-t001:** Comparison of hythane yields with other relative works.

Substrates	Pretreatment Method	Conditions	Microorganism	T, °C	YH_2_	YCH_4_	YHythane	References
Rice straw	Mechanical crushing under 2 mm. SS and rice straw codigestion	B	SS	55	21 mL/g VS	266 mL/g VS	287 mL/g VS	[32]
Corn stalk	0.5% H_2_SO_4_ and 10% *w*/*w* at 121 °C for 60 min	B	*Bacillus* sp. FS2011and CDS	55	88.1 mL/g VS	227 mL/g COD	306.8 mL/g COD	[33]
Cornstalk	Hydrothermal liquefaction, 260 °C for 0 min, vacuum filtration	UASB to PBR	SS	37	146 mL H_2_/g COD	302 mL CH_4_/g COD	448 mL/g COD	[17]
Wheat straw hydrolysate	Provided by Risø DTU (Denmark)	UASB to UASB	Mixed-culture and methonogenic granules and BS	55 and 70	89 mL/g VS	307 mL/g VS	396 mL/g VS	[8]
Rice straw enzymatic hydrolyzate	NU pretreatment at outdoor winter and 100% *w*/*v* for 3 months	B	Thermoanaerobacterium thermosaccharolyticum W16 and BS	60	155.5 mL/g VS	63.1 mL/g VS	218.6 mL/g VS	[13]
Rice straw	NU pretreatment at outdoor winter and 100% *w*/*v* for 3 months	IABR	BS	55–60	104.1 mL/g VS	508.4 mL/g VS	612.5 mL/g VS	This research

B is shown as batch; CDS is cow dung sludge; AS is anaerobic sludge; SS is sewage sludge; BS is biogas slurry.

## Data Availability

The data presented in this study are available on request from the corresponding author.

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
