# Peer review of "Improved Biohythane Production from Rice Straw in an Integrated Anaerobic Bioreactor under Thermophilic Conditions"

_microorganisms, 2023, doi:10.3390/microorganisms11020474_

Round 1
Reviewer 1 Report
This study evaluated the feasibility of continuously biohythane production from rice 11 straw (RS) using an integrated anaerobic bioreactor (IABR) at thermophilic condition. Some minor issues should be addressed before publication:
(1) Page 3, Line 182-184, "It is reported that substrate concentration is one of the key factors affecting the productivity of a bioenergy generating system, especially biohydrogen production process", the authors are suggested to explain more detaily;
(2) Page 3, Line 194-196,"This could be also explained NU pretreat-194 ment at outdoor cold-winter conditions showed a practical and feasible way for improving biodegradability of lignocellulose." the authors are suggested to explain how to improve biodegradability of lignocellulose by the pretreatment. Physical or chemicals?
overall, the paper is well organised and can be accepted after minor reversion.
Author Response
To reviewer #1
Dear Reviewer:
Thank you very much for your letter and the comments regarding our paper submitted to Microorganisms. We have carefully considered the comments and have revised the manuscript accordingly. We submit here the revised manuscript as well as a list of changes.
- Comment:Page 3, Line 182-184, “It is reported that substrate concentration is one of the key factors affecting the productivity of a bioenergy generating system, especially biohydrogen production process”, the authors are suggested to explain more detaily;
Response: We sincerely appreciate the reviewer’s suggestions. We have added more details and marked them in yellow in the manuscript.
- Comment:Page 3, Line 194-196, “This could be also explained NU pretreat-194 ment at outdoor cold-winter conditions showed a practical and feasible way for improving biodegradability of lignocellulose.” the authors are suggested to explain how to improve biodegradability of lignocellulose by the pretreatment. physical or chemicals?
Response: Thanks for the reviewer’s valuable suggestion. We have added more details about NU pretreatment and marked them in yellow in the manuscript. What’s more, NU pretreatment has been carried out comprehensive and in-depth research in our previous studies (Dong, L.L.; Cao, G.L.; Zhao, L.; Liu, B.F.; Ren, N.Q. Alkali/urea pretreatment of rice straw at low temperature for enhanced biological hydrogen production. Bioresour. Technol 2018. 267, 71-76.; Dong, L.L.; Cao, G.L.; Wu, J.W., Liu, B.F., Xing, D.F., Zhao, L., Zhou, C.S., Feng, L.P., Ren, N.Q. High-solid pretreatment of rice straw at cold temperature using NaOH/Urea for enhanced enzymatic conversion and hydrogen production. Bioresour. Technol 2019. 287, 121399.; Dong, L.L.; Wu, J.W., Zhou, C.S., Xu, C.J., Liu, B.F., Xing, D.F., Xie, G.J., Wu, X.K., Wang, Q., Cao, G.L.; Ren, N.Q. Low concentration of NaOH/Urea pretreated rice straw at low temperature for enhanced hydrogen production. Int J Hydrogen Energ, 2020. 45, 1578-1587.; Dong, L.L.; Wu, X.K.; Wang, Q.; Cao, G.L.; Wu, J.W.; Zhou, C.S.; Ren, N.Q. Evaluation of a novel pretreatment of NaOH/Urea at outdoor cold-winter conditions for enhanced enzymatic conversion and hythane production from rice straw. Sci. Total. Environ 2020. 744, 140900.). Previous studies showed that NU pretreatment operated directly at outdoor in cold winter and at a higher solid loading was a feasible technology for enhancing energy production from lignocellulose. Thus, the main aim of this study was to focus on continuous biohythane production from rice straw directly. Also, this study is an extension of previous research.
Besides the above changes, we have examined the paper carefully and corrected some expression errors. These changes will not influence the content and framework of the manuscript. And here, we did not list the changes but marked them in yellow in the revised paper. We appreciate Reviewer’s warm work earnestly and hope that the correction will meet with approval.
Special thanks to you for your good comments.
Best regards,
Yours sincerely,
Li-li Dong; E-mail: donglili0569@126.com; Fax: +86 898 66269468
Guang-Li Cao; E-mail: caogl@hit.edu.cn; Fax: +86 451 86402652
Reviewer 2 Report
The article is aimed at continuously biohythane production from rice straw. To implement the process, it is proposed to use an integrated thermophilic anaerobic bioreactor. Obtaining energy carriers with the help of anaerobic bioconversion is an extremely important and urgent task, since the production of waste annually increases with a simultaneous increase in energy consumption. The article looks complete, but there are some questions about the design and content, so I recommend the major revision.
Key comments:
1. In the introduction, it is stated that hytane is a mixture of hydrogen and methane. From the content of the article, it can be concluded that the authors believe that the ratio of hydrogen and methane in githane or biogytan can be any, but there are certain ratios. This should be reflected in the introduction.
2. Lines 62-74 from the introduction should be moved to discussions.
3. Section 2.1 lists characteristics for inoculum (COD, VS, TS etc.), but not for feedstock. It is extremely important to indicate them, because the results obtained are questionable.
4. Figure 2a shows 6 curves, but the legend shows 3.
5. Figure 2 is difficult to understand - it is not clear what the authors want to show them.
6. On line 177, the dimensions L/L are indicated, referring to Figure 2, in which this dimension is not present. In figure 2, you should indicate not l (ml), but L / L.
7. In figure 3, L/L should also be indicated, since the production of biogas in ml is misleading and the reader needs to look for the volume of the bioreactor.
8. It is not clear why it was necessary to complicate Figure 3a by introducing an auxiliary axis, if a constant HRT was maintained throughout almost the entire experiment, except for the start of the process. Figure 3b is also complicated: why is the carbon dioxide content indicated?
9. It is not clear for what purpose in Section 3.2.2 the specific yields of biogases per g TS are given. This is completely uninformative and should be corrected for g VS or COD. At the same time, as indicated in comment 3, these characteristics are not given in the article. In this regard, it is absolutely not clear how much biogas is obtained in the described process.
10. In figure 4, you should also indicate L / L.
11. In figure 6, for clarity, specific dimensions should be indicated.
12. Table 1 needs to be corrected according to the comments (ml/g VS or COD).
13. The equation (line 354) should be numbered and placed in section 2.
14. Accordingly, the abstract and conclusion should be revised indicating the values obtained in terms of ml/g VS or COD.
Author Response
To reviewer #2
Dear Reviewer:
Thank you very much for your letter and the comments regarding our paper submitted to Microorganisms. We have carefully considered the comments and have revised the manuscript accordingly. We submit here the revised manuscript as well as a list of changes.
- Comment: In the introduction, it is stated that hytane is a mixture of hydrogen and From the content of the article, it can be concluded that the authors believe that the ratio of hydrogen and methane in githane or biogytan can be any, but there are certain ratios. This should be reflected in the introduction.
Response: We sincerely appreciate the reviewer’s suggestions. We have added the ratio of H2/CH4 in the introduction. A mixture of H2 and CH4 is called hythane, HCNG or methagen. Usually, H2 content in hythane is 5 - 25% by volume. Typically, the suggested H2 content in hythane is 10 - 25% by volume. By combining the advantages of H2 and CH4, hythane is considered one of the important fuels involved in achieving the transition of technical models from a fossil fuel-based society to a terminal hydrogen-based society. Moreover, it has been reported that the higher the ratio of H2/CH4, the better the quality of the hythane.
- Comment: Lines 62-74 from the introduction should be moved to discussions.
Response: As suggested, we have moved this part (Lines 62-74) to discussions and marked them in yellow in the manuscript.
- Comment:Section 2.1 lists characteristics for inoculum (COD, VS, TS etc.), but not for feedstock. It is extremely important to indicate them, because the results obtained are questionable.
Response: We sincerely appreciate the reviewer’s suggestions, we have listed characteristics for feedstock in Section 2.1 and marked them in yellow in the manuscript.
- Comment:Figure 2a shows 6 curves, but the legend shows 3.
Response: We are sorry that the uncleared expression confuses the reviewer. We have redrawn Figure 2a.
- Comment:Figure 2 is difficult to understand - it is not clear what the authors want to show them.
Response: We are sorry that the uncleared expression confuses the reviewer. We have redrawn Figure 2.
- Comment:On line 177, the dimensions L/L are indicated, referring to Figure 2, in which this dimension is not present. In figure 2, you should indicate not l (ml), but L / L.
Response: As suggested, we have replaced “mL” with “L/L” in figure 2.
- Comment:In figure 3, L/L should also be indicated, since the production of biogas in ml is misleading and the reader needs to look for the volume of the bioreactor.
Response: As suggested, we have replaced “mL” with “L” in figure 3 and redrawn figure 3.
- Comment: It is not clear why it was necessary to complicate Figure 3a by introducing an auxiliary axis, if a constant HRT was maintained throughout almost the entire experiment, except for the start of the process. Figure 3b is also complicated: why is the carbon dioxide content indicated?
Response: We are sorry that the uncleared expression confuses the reviewer. We have complicated Figure 3a and redrawn Figure 3a. In this study, hydrogen (H2) and carbon dioxide (CO2) were the main products of HPA, and methane (CH4) and CO2 in MPA. Meanwhile, the basis of carbon content in gas was provided to calculate of carbon distribution in 3.3. Thus, the carbon dioxide content was presented in Figure 3b.
- Comment: It is not clear for what purpose in Section 3.2.2 the specific yields of biogases per g TS are given. This is completely uninformative and should be corrected for g VS or COD. At the same time, as indicated in comment 3, these characteristics are not given in the article. In this regard, it is absolutely not clear how much biogas is obtained in the described process.
Response: We are sorry that the uncleared expression confuses the reviewer. We have rewritten this portion and corrected it for g VS. At the same time, we have carefully checked the whole Section 3.2.2 according to the reviewer’s suggestion to avoid similar expressions.
- Comment: In figure 4, you should also indicate L/L.
Response: As suggested, we have redrawn figure 4.
- Comment: In figure 6, for clarity, specific dimensions should be indicated.
Response: Thanks for the reviewer’s valuable suggestion, we have indicated the specific dimensions of reactor and redrawn figure 6.
- Comment: Table 1 needs to be corrected according to the comments (ml/g VS or COD).
Response: As suggested, we have corrected Table 1, the unit is mL/g VS.
- Comment: The equation (line 354) should be numbered and placed in section 2.
Response: As suggested, we have placed the equation (line 354) in section 2. Moreover, we have also rewritten the relevant section and marked them in yellow in the manuscript.
- Comment: Accordingly, the abstract and conclusion should be revised indicating the values obtained in terms of ml/g VS or COD.
Response: We sincerely appreciate the reviewer’s suggestions. We have revised mL/g VS in the abstract and conclusion and marked them in yellow in the manuscript.
Besides the above changes, we have scrutinized the manuscript and corrected some expression errors. These changes will not influence the content and framework of the manuscript. And here we did not list the changes but marked them in yellow in revised paper. We appreciate for Reviewer’s warm work earnestly and hope that the correction will meet with approval.
Special thanks to you for your good comments.
Best regards,
Yours sincerely,
Li-li Dong; E-mail: donglili0569@126.com; Fax: +86 898 66269468
Guang-Li Cao; E-mail: caogl@hit.edu.cn; Fax: +86 451 86402652
Reviewer 3 Report
In this work, Dong et al. have investigated the feasibility of continuous biohythane production from rice straw using an integrated anaerobic bioreactor. Their results demonstrate that pretreatment via NaOH/Urea solution can enhance H2 production and hythane quality. The energy conversion efficiency of continuous biohythane production is significantly higher than single H2 and CH4 production process. This work is of interest to the audience of Microorganisms. There are few concerns, though, that need to be addressed before acceptance.
1. Ln 16: COD is not defined until in Ln 110.
2. Ln 49-51: Please check the grammar and rephrase this sentence.
3. Ln 52: “pretreatment are needed”
4. Ln 62: UASB was not defined anywhere.
5. Ln73: CSTR was not defined anywhere.
6. Ln 119: the before and after pretreatment of RS was used ïƒ the RS before and after pretreatment was used
7. Figure 2a is confusing. Why there are six curves in Figure 2a?
8. Figure 2b: a typo in the y-axis label “Ggas”
9. The unit of gas production is % in Figure 2d, whereas this unit is ml in Figure 2a and 2c.
10. Figure 2b-2d: The color of the CH4 production profiles does not match the figure legend.
11. Is that necessary to plot both H2/CH4 content (Figure 2b) and H2/CH4 production (Figure 2c)? I would suggest the authors to reorganize Figure 2 to compare the pretreated rice straw and un-pretreated rice straw side by side.
12. Figure 3b: a typo “CH2”
13. Figure 4: Please use the same color to represent H2 or CH4 production, i.e., use green squares for H2 (closed squares for H2 and open squares for specific H2) and orange circles for CH4 (closed circles for CH4 and open circles for specific CH4).
14. Table 1 needs improvement. Either reduces the font size or rotate the whole Table to be vertical.
Author Response
To reviewer #3
Dear Reviewer:
Thank you very much for your letter and the comments regarding our paper submitted to Microorganisms. We have carefully considered the comments and have revised the manuscript accordingly. We submit here the revised manuscript as well as a list of changes.
- Comment:Ln 16: COD is not defined until in Ln 110.
Response: Thanks for the reviewer’s suggestion. We have defined COD and marked it in yellow in the manuscript.
- Comment:Ln 49-51: Please check the grammar and rephrase this sentence.
Response: Thanks for the reviewer’s suggestion. We have rephrased this sentence and revised the whole manuscript carefully according to the reviewer’s request to avoid grammar errors. In addition, we asked colleagues experts in English to help edit the manuscript. We believe that the language is now acceptable for the journal.
- Comment:Ln 52: “pretreatment are needed”.
Response: As suggested, we have added “pretreatment” in this sentence and marked it in yellow in the manuscript.
- Comment:Ln 62: UASB was not defined anywhere.
Response: As suggested, we have defined UASB and marked it in yellow in the manuscript.
- Comment:Ln73: CSTR was not defined anywhere.
Response: As suggested, we have defined CSTR and marked it in yellow in the manuscript.
- Comment:Ln 119: the before and after pretreatment of RS was used the RS before and after pretreatment was used.
Response: Thanks for the reviewer’s suggestion. We have replaced “the before and after pretreatment of RS was used” with “the RS before and after pretreatment was used” and marked them in yellow in the manuscript.
- Comment:Figure 2a is confusing. Why there are six curves in Figure 2a?
Response: We are sorry that the uncleared expression confuses the reviewer. We have redrawn Figure 2.
- Comment:Figure 2b: a typo in the y-axis label “Ggas”.
Response: We are sorry that the uncleared expression confuses the reviewer. We have redrawn Figure 2. Moreover, we have revised the whole figures carefully according to the reviewer’s suggestion to avoid typo errors.
- Comment:The unit of gas production is % in Figure 2d, whereas this unit is ml in Figure 2a and 2c.
Response: We are sorry that the uncleared expression confuses the reviewer. We have redrawn Figure 2.
- Comment:Figure 2b-2d: The color of the CH4 production profiles does not match the figure legend.
Response: Thanks for the reviewer’s suggestion. We have changed the color of the CH4 production profiles and made it match the figure legend.
- Comment:Is that necessary to plot both H2/CH4 content (Figure 2b) and H2/CH4 production (Figure 2c)? I would suggest the authors to reorganize Figure 2 to compare the pretreated rice straw and un-pretreated rice straw side by side.
Response: We sincerely appreciate the reviewer’s suggestions. Indeed, the ratio of H2/CH4 was an essential parameter in this study. Moreover, the main aim of 3.1 was to realize and obtain the effect of substrate concentration and HRT for IABR operation. Meanwhile, we didn’t give this parameter and other information in this figure to avoid confusing the reviewer. However, as suggested, we have reorganized Figure 2.
- Comment:Figure 3b: a typo “CH2”.
Response: We are sorry that a wrong typo confuses the reviewer. We have replaced “CH2” with “CH4”.
- Comment:Figure 4: Please use the same color to represent H2 or CH4 production, i.e., use green squares for H2 (closed squares for H2 and open squares for specific H2) and orange circles for CH4 (closed circles for CH4 and open circles for specific CH4).
Response: As suggested, we have reorganized Figure 4.
- Comment:Table 1 needs improvement. Either reduces the font size or rotate the whole Table to be vertical.
Response: We sincerely appreciate the reviewer’s suggestions. We have reorganized and supplied more details in Table 1.
Besides the above changes, we have scrutinized the paper carefully and corrected some expression errors. These changes will not influence the content and framework of the manuscript. And here, we did not list the changes but marked them in yellow in revised paper. We appreciate Reviewer’s warm work earnestly and hope that the correction will meet with approval.
Special thanks to you for your good comments.
Best regards,
Yours sincerely,
Li-li Dong; E-mail: donglili0569@126.com; Fax: +86 898 66269468
Guang-Li Cao; E-mail: caogl@hit.edu.cn; Fax: +86 451 86402652
Round 2
Reviewer 2 Report
Good work has been done to eliminate the shortcomings. However, the main remark was not eliminated. It is well known that 350 ml of methane can be obtained from 1 gram of decomposed COD. In this work, on lines 190-191 it is indicated that 1 gram of VS of this substrate is approximately equal to 1 gram of COD. At the same time, in addition to 508.4 ml of methane, an additional 104.1 ml of hydrogen was obtained from one gram of COD. Thus, either the authors made serious errors in the calculation, or they are deliberately trying to mislead.
Key comments:
1. Lines 289-290 indicate that Figure 3a illustrates the relationship between daily gas production and OLR. Neither this nor the previous version shows this relationship.
2. OLR is not listed at all in the article, despite the fact that it is one of the most important indicators of semi-continuous fermentation.
3. Gas (hythane, hydrogen, methane) production rate has the dimension L/(L day)
4. From the given material it is not clear what was the hydraulic retention time in MPA: 13 or 10 days. Did the reactor operate in thermophilic mode?
5. In the figures and in the text, it is necessary to indicate gases production rate (L/(Lday)) and their specific yields (ml/g VS) for semi-continuous operation.
6. It is necessary to indicate under what conditions (OLR, HRT, temperature) the reactor operated.
Author Response
To reviewer #2
Dear Reviewer:
Thank you very much for your letter and the comments regarding our paper (microorganisms-2146767) submitted to Microorganisms. We have carefully considered the comments and have revised the manuscript accordingly. We submit here the revised manuscript as well as a list of changes.
Point 1: Lines 289-290 indicate that Figure 3a illustrates the relationship between daily gas production and OLR. Neither this nor the previous version shows this relationship.
Response 1: We are sorry that the wrong expression confuses the reviewer. In the first submitted manuscript, we show the OLR information in Figure 3a. However, the reviewer’s suggestion OLR information should be deleted, due to a constant OLR being maintained throughout almost the entire experiment, except for the start of the process. We adopted the reviewer’s suggestion and reorganized Figure 3. However, we made a mistake, and the manuscript’s expression was not modified. In this revised manuscript, we have changed the relevant phrases and marked them in yellow.
Point 2: OLR is not listed at all in the article, despite the fact that it is one of the most important indicators of semi-continuous fermentation.
Response 2: We sincerely appreciate the reviewer’s suggestions. We have provided more details about OLR in Section 2.3.2 and marked them in yellow.
Point 3: Gas (hythane, hydrogen, methane) production rate has the dimension L/(L day).
Response 3: Thanks for the reviewer’s valuable suggestion. We have unified gas (hythane, hydrogen, methane) production rate (L/(L day)). Moreover, we have redrawn Figure 2.
Point 4: From the given material it is not clear what was the hydraulic retention time in MPA: 13 or 10 days. Did the reactor operate in thermophilic mode?
Response 4: We are sorry that the uncleared expression confuses the reviewer. The hydraulic retention time (HRT) in MPA is 10 days. You understand correctly. The reactor operates in thermophilic mode (60 °C). The external IABR was surrounded by a tropical belt to keep the constant temperature required for the bioreactor. The expressions were modified and provided in the article.
Point 5: In the figures and in the text, it is necessary to indicate gases production rate (L/(Lday)) and their specific yields (ml/g VS) for semi-continuous operation.
Response 5: We sincerely appreciate the reviewer’s suggestions. We have provided gases production rate (L/(Lday)) and their specific yields (mL/g VS) for batch experiments and marked them in yellow.
Point 6: It is necessary to indicate under what conditions (OLR, HRT, temperature) the reactor operated.
Response 6: We sincerely appreciate the reviewer’s suggestions. We have provided more details about the conditions the reactor operated in Section 2.3.2 and marked them in yellow.
Point 7: It is well known that 350 ml of methane can be obtained from 1 gram of decomposed COD. In this work, on lines 190-191 it is indicated that 1 gram of VS of this substrate is approximately equal to 1 gram of COD. At the same time, in addition to 508.4 ml of methane, an additional 104.1 ml of hydrogen was obtained from one gram of COD. Thus, either the authors made serious errors in the calculation, or they are deliberately trying to mislead.
Response 7: We sincerely appreciate the reviewer’s comments. Indeed, 350 ml of methane can be obtained from 1 gram of decomposed COD. However, in this work, 1% VS of this substrate concentration is approximately equal to about 10000mg COD/L culture, also 10 gram/L of COD, isn’t 1 gram of COD. The specific biohythane yield was shown by mL/g VS, because the carbon in RS was transformed into carbon-containing metabolites in HPA (soluble polysaccharides, VFAs) via hydrolytic acidification process and methane. In another word, the carbon source was derived from the continuous hydrolysis of solid matter in HPA. Then, the liquid metabolites and residual solids of HPA could serve as the carbon source of MPA for CH4 production. Thus, the total amount of organic matter is certain when it is influnted, and the outflunt also is certain. The difference is that the process is always accompanied by the formation of gas associated with transforming the solid phase into the liquid phase. Therefore, the specific biohythane yield was shown by mL/g VS.

Round 3
Reviewer 2 Report
I thank the authors for the work done on the comments. In general, the article can be published after rechecking the dimensions and other typographical errors.
1. On line 144, the value of OLR is indicated, but its dimension should be g / (L day)
2. On lines 177-178, as well as on line 200, the gases production rate values are indicated, which are two to three orders of magnitude higher than those of existing installations. Please check the dimension .
Author Response
To reviewer #2
Dear Reviewer:
Thank you very much for your letter and the comments regarding our paper (microorganisms-2146767) submitted to Microorganisms. We have carefully considered the comments and have revised the manuscript accordingly. We submit here the revised manuscript as well as a list of changes.
Point 1: On line 144, the value of OLR is indicated, but its dimension should be g / (L day).
Response 1: Thanks for the reviewer’s valuable suggestion. We have changed the value of OLR to g/(L day).
Point 2: On lines 177-178, as well as on line 200, the gases production rate values are indicated, which are two to three orders of magnitude higher than those of existing installations. Please check the dimension .
Response 2: We are sorry that the wrong expression confuses the Reviewer. We have changed the gas production rate values and checked the whole manuscript. Moreover, we have provided the H2 production rate values in the manuscript and marked them in yellow.
